# Concealed for a Long Time on the Marches of Empires: Hepatitis B Virus Genotype I

**DOI:** 10.3390/microorganisms11092204

**Published:** 2023-08-31

**Authors:** Agnès Marchio, Philavanh Sitbounlang, Eric Deharo, Phimpha Paboriboune, Pascal Pineau

**Affiliations:** 1Institut Pasteur, Université Paris Cité, Unité “Organisation Nucléaire et Oncogenèse”, INSERM U993, 75015 Paris, France; agnes.marchio@pasteur.fr; 2Centre d’Infectiologie Lao-Christophe Mérieux (CILM), Vientiane 3888, Laos; philavanhsbl@gmail.com (P.S.); phimpha@cilm-laos.org (P.P.); 3MIVEGEC, Université Montpellier, CNRS, IRD, 34394 Montpellier, France; eric.deharo@ird.fr

**Keywords:** *Hepadnaviridae*, HBV, genotype I, recombinant, ethnic minorities, Southeast Asian Massif, Zomia, Guangxi

## Abstract

Genotype I, the penultimate HBV genotype to date, was granted the status of a bona fide genotype only in the XXIst century after some hesitations. The reason for these hesitations was that genotype I is a complex recombinant virus formed with segments from three original genotypes, A, C, and G. It was estimated that genotype I is responsible for only an infinitesimal fraction (<1.0%) of the chronic HBV infection burden worldwide. Furthermore, most probably due to its recent discovery and rarity, the natural history of infection with genotype I is poorly known in comparison with those of genotypes B or C that predominate in their area of circulation. Overall, genotype I is a minor genotype infecting ethnic minorities. It is endemic to the Southeast Asian Massif or Eastern Zomia, a vast mountainous or hilly region of 2.5 million km^2^ spreading from Eastern India to China, inhabited by a little more than 100 million persons belonging primarily to ethnic minorities speaking various types of languages (Tibeto-Burman, Austroasiatic, and Tai-Kadai) who managed to escape the authority of central states during historical times. Genotype I consists of two subtypes: I1, present in China, Laos, Thailand, and Vietnam; and I2, encountered in India, Laos, and Vietnam.

## 1. Introduction

In a group that counts ten human-infecting members from A to J, the penultimate hepatitis B virus (HBV) representative, genotype I, discovered only on the eve of the XXIst century, occupies the uncomfortable position of an illegitimate child [1,2,3]. Genotype I is apparently built up of three DNA segments originally found in genotypes C, G, and A, by order of heritage importance [4,5]. At first, it was, thus, difficult for some of the most influential specialists of the HBV to consider genotype I as a bona fide member of the family [6]. In addition, although it is present in a vast and populated region where chronic infection with the HBV used to be highly endemic, genotype I occupies a very small niche, corresponding, when it is present, to only a small fraction of all HBV infections diagnosed locally [1,7]. Furthermore, although some of the nations concerned by its presence, such as China or India, are immense, in both, genotype I tends to preferentially infect ethnic minorities living in border areas distant from the central power, thereby reinforcing its marginal status [8,9]. Finally, the clinical and biological properties of genotype I are poorly documented and might, fortunately, remain so in the era of massive anti-hepatitis B immunization at birth. In the following review, we shall expose the major molecular features of genotype I and describe some of the characteristics of the populations exposed to it.

## 2. Virus Discovery

Genotype I was discovered at the end of the XXth century by the group of Lindh and Norder, who analyzed “aberrant” strains of the hepatitis B virus (HBV) isolated from five patients living around Hanoï, the capital of Vietnam. At that time, Hannoun et al. correctly identified the genetic blocks originating from genotypes C and A, which are the true components of genotype I [10]. However, the discovery of genotype G in the same year (i.e., 2000) was too recent, and the corresponding sequences were too scarce to enable them to easily reconstruct the full genotype I puzzle [11]. Consequently, and understandably, they neither dared to be assertive regarding the existence of a new HBV genotype nor to firmly name its original components. Eight years later, two groups working in Vietnam and Laos came back on the track of the “aberrant HBV strains” that they will name genotype I [4,5]. In the interval, the work of Peter Simmonds shed light on the frequent inter- and intra-genotype recombination events affecting the HBV [12]. The availability of more sequences from genotype G and the conceptual framework for recombination events in Hepadnaviridae enables both Huy and coworkers and Olinger and colleagues to propose, five months apart from one another in 2008, genotype I as the ninth genotype of the HBV. Huy et al. were originally working on a single isolate from a patient living in Hanoi (Vietnam) to whom they aggregated the Vietnamese strains published in Sweden in 2000, while Olinger et al. was analyzing blood samples from first-time blood donors in Vientiane, Laos. Both teams consistently recognized the novel and chimeric nature of genotype I formed from three genomic blocks originating from different genotypes, C, G, and A, by decreasing order of their DNA amount borrowed by genotype I (Figure 1A,B). Olinger and colleagues, who had at their disposal more sequences (n = 19, including 15 complete genomes), were even able to propose the existence of two subtypes I1 and I2. To be perfectly fair, at the same time, Indian investigators began to figure out that HBV strains circulating in a small mountain area from an Eastern Indian state, Arunachal Pradesh, were untypable. These isolates were proven to be genotype I later on [13]. The claim for the creation of a new genotype immediately found opposition among scientific community leaders due to the fact that genotype I is not an original genotype but an assembly of three previously known genotypes [6]. However, the International Committee on Taxonomy of Virus (ICTV) consulted in June 2023 included as an independent genotype an I strain in its phylogeny (https://ictv.global/report/chapter/hepadnaviridae/hepadnaviridae, accessed on 1 July 2023).

## 3. Genotype I as a True Genotype

The genome size of HBV genotype I is 3215 nucleotides, as with four other genotypes (B, C, F, and H) [2]. As mentioned above, genotype I, therefore, represents the stable and successful outcome of recombination events, allowing its classification as a bona fide genotype. Recombination events are, by definition, restricted to the patient where they occur, then to a cluster of individuals belonging to the same family, notably in Eastern Asia, where pseudo-vertical or perinatal transmissions from mothers to newborns are very frequent [14]. The status of genotype I contrasts with other stable recombinant forms of the HBV that are commonly transmitted and successfully spread in specific populations, albeit without reaching so far as to attain the status of a genotype. It is the case of the genetically and geographically well-defined C/D recombinant that predominates in Tibetan populations [15]. However, the genetic distance that separates genotype I from its closest ancestor, genotype C, is 7.5%, suggesting that it deserves to be considered as a true genotype [5]. In addition, taking diversity as a hallmark of seniority in the family of HBV genotypes, it appears that genotype I is probably not the most recent Hepadnavirus that successfully spread in human populations. Olinger and coworkers have suggested in their 2008 paper that Laotian HBV genotype I isolates could be consistently divided into two branches, presenting a maximal genetic distance of 4.3% for this group of full sequence, despite the average distance between all genotype I isolates being only 2.3% (above that observed in genotypes E, G, and H, however). In addition, Laotian subtypes I1 and I2, proposed by Olinger et al., belonged to different serotypes (ayw1 and adw2), thereby strengthening the biological relevance of these subtypes within the genotype I taxon (Figure 2) [5,16,17].

## 4. Genome Architecture

The genome of HBV genotype I is, thus, assembled from three pieces taken from three different ancestral genotypes: A, C, and G [4,5] (Figure 1A,B). The largest part, which can be considered the backbone of the virus, was composed of a genotype C strain; it extends grossly from position 1400 to 3000 of the genome (50%). The second component in importance consists of genotype G, which covers a region approximately extending from nucleotide 400 to 1400. Finally, the last and smallest inheritance of genotype I is from genotype A, between positions 3000 and 400 of the circular genome that counts, as already mentioned, 3215 nucleotides. Of course, the boundaries of the three components that result from recombination events cannot be determined with nucleotide-wise precision, and recombinant detection software such as SimPlot (downloadable for free upon request from Dr. Stuart Ray from the Dept of Medicine, Johns Hopkins University School of Medicine, email: sray@jhmi.edu) or jpHMM (freely available at http://jphmm.gobics.de/, accessed on 1 July 2023) are often hesitant around recombination boundaries, as observable by the steep decrease of prediction confidence on the original fragments’ boundaries [18,19]. The proportions of te ancestral genotypes for each HBV genotype I gene and product are given in Table 1. Concerning the overall nucleotide diversity of genotype I, Phan and coworkers observed a lower amount of diversity here than in other Australasian HBV genotypes (A, B, C, and D) except between nucleotides 2500–2600 (terminal protein domain) [20].

## 5. Hypothetical Genesis

Although genotypes A and G can be found sporadically in Eastern Asia, they represent infrequent encounters for virologists in China, Laos, or Vietnam [21,22,23]. This situation is probably responsible for the initial perplexity of Chinese investigators confronted with genotype I, which might explain how this genotype was, at first, designated as an X/C (X for unknown/anonymous) recombinant in China [9,24]. The presence of large DNA segments apparently originating from two non-autochthonous viruses in a genotype itself endemic to southeastern continental Asia can, at least, be considered surprising.

The fact that genotype I results from the successful recombination of three genotypes is also amazing. Recombinants formed with two genotypes are frequent but tripartite associations less so [25]. Thus, the molecular and geographic origins of genotype I are a legitimate concern. The last enigma concerns, of course, the G segment of genotype I. The countries where genotype G is the most abundant are all located in the Americas, from Mexico to Argentina through Colombia [1]. Genotype G is conspicuously absent from Eastern Asia but is sporadically found in the mixed populations (France, Italy, The Netherlands, Spain, and Switzerland) living in Western Europe at the other extremity of the Eurasian block. 

HBV infections with mixtures of genotypes are frequent, especially in HIV-infected patients or in regions where the rate of chronic infection is high and the pool of infected individuals is large, as in China [26,27]. However, due to the relative infrequency of triple recombinants, it is not the most plausible hypothesis to consider that genotype I was generated at once by at least two recombination events occurring in a single patient with a triple-genotype infection. Genotype I is more probably the result of sequential recombination events for which the order and exact partnership of the component genotypes are unknown (A/G then C, A/C then G, or A/G then C). Due to the molecular and biological characteristics of genotype G, which presents two stop codons in HBeAg and, thus, is incapable of installing on its own an immune tolerance in its hosts, we consider that genotype G was the primary target in the first recombination event [28,29]. To allow the “survival” of the recombinant virus, this event should have replaced the genotype G HBe gene with a foreign HBe protein-producing gene. Such recombinants have been described on several occasions [25]. The next question is knowing what the first partner of this genotype G strain was. An examination of the geographical distribution of genotype G and its candidate bedfellows in genotype I, i.e., genotypes A or C, favors the hypothesis of genotype A as the initial partner. Indeed, A/G recombinants are rather frequent in nucleotide databases, perhaps more frequent than they should be with regard to the rarity of genotype G. Furthermore, a significant proportion of the A/G recombinant removes, as expected, the immunologically problematic HBe region of genotype G in the recombined progeny, and a certain number of them (GeneBank accessions HM484898, HM484906, and HM48492) are compatible with the formation of genotype I (Figure 3) [30]. The final molecular event that led to the generation of the new genotype I must be, therefore, a recombination between this putative ancestral A/G recombinant with a representative genotype C (Figure 3). This recombination event removed, for the second time, the HBe region to replace the genotype A version with the more efficient genotype C version. It is well-known that genotype C plays the role of a major provider of this particular genome segment to other Asian genotypes, as illustrated by subtypes B2–B5, which contain all of them: a core promoter, a pre-core, and a core genomic region taken from a genotype C strain (Figure 4 and Table 1) [31]. Of course, we admit that this two-step scenario is purely speculative, and a deeper and more skilled analysis of genotype I sequences and of its putative two-partner ancestors might invalidate our hypothesis.

An assessment of the homologies of the genotype I genome with those of other genotypes indicates clearly that it is more related to subtypes C6 and C12, present in Indonesia, than with other C subtypes (our observations). The homologies with current subtypes of genotype A are less clear. Interestingly, some authors still consider that the non-C part of genotype I pertains to an unknown genotype or does not validate the presence of genotype A in this recombinant [32,33]. Importantly, genotype I itself does not escape the process that allowed for its emergence, as B/I or C/I recombinants have already been described in China and Laos [22,34,35].

## 6. Current Geographic Distribution

According to Velkov and coworkers, 94.0% of HBV genotype I infections occur in Eastern Asia. And its contribution to the global burden of chronic HBV infections is 0.32% [1]. More precisely, genotype I is endemic to eastern continental Asia, as it was not found so far in the insular countries of the area (Japan, Taiwan, Philippines, Malaysia, Indonesia, Brunei, and Timor Leste) [1]. Currently, its north-to-south distribution spreads from Northwestern China (Xibei, including Shaanxi province) to Thailand and Vietnam, and from west to east from Arunachal Pradesh in East India to the Guangxi autonomous region in South Central (Zhongnan) China [4,5,8,9,21,36,37,38]. In practice, genotype I was found in five countries: Vietnam, Laos, India, China, and Thailand (Figure 5). In China, the presence of genotype I was reported from Sichuan, Shaanxi, Yunnan, and Guangxi, while, in India, it was found only in Arunachal Pradesh. Overall, genotype I primarily infects populations living in inland mountainous or hilly regions, including the Eastern Himalayas. As an illustration, in Thailand, genotype I was found in Chiang Mai, north of the country, while, in Vietnam, genotype I was isolated in patients living around Hanoi in the north but was absent from a genotyping survey conducted on 135 HBV carriers in Ho Chi Minh City, southern Vietnam [4,10,23]. In Laos, genotype I was found in blood donors from the national capital Vientiane, located in the northern part of the country [22]. We isolated five genotype I strains in chronically infected patients from regions (Vientiane, Xiengkouang, Huaphan, and Phongsaly) located in Northern Laos but none from Central and South Laos [39]. Outside Southeast Asia, two patients both originating from Vietnam have been characterized in France and Canada [40,41].

Genotype I seems to be completely absent from the regions and countries flanking this ensemble. It has not been found in Myanmar so far, despite common borders with four territories where genotype I is present [42]. Its detection is neither reported in Bhutan, the western neighbor of Arunachal Pradesh, nor in Indian states neighboring Arunachal Pradesh (Assam, Nagaland, etc.), suggesting it is endemic to a single population [43]. However, molecular studies from Myanmar of Bhutan are quite rare. It is unknown if it infects autochthonous coastal populations from China, Vietnam, or Thailand or if its distribution area is restricted, as it seems, to populations living in intra-continental mountain areas within these countries. 

Finally, subtypes I1 and I2 do not share the same distribution areas. Subtype I1 is endemic in South China, Vietnam, and Laos, while Subtype I2 is present in Laos and India (Figure 5).

## 7. Importance of HBV Genotype I Infections in Its Endemic Area

In this vast region where chronic infection by the HBV used to be highly endemic (>8.0% of HBsAg(+) in the population) and tends to be moderately high (5.0–8.0%) nowadays, genotype I is generally present in a minor fraction of HBV carriers [44]. 

As an example, in Laos, blood donors presenting an infection with genotype I represents 2.4% of HBV infections [5]. As already mentioned, genotype I is present in a state of Northeastern India, Arunachal Pradesh, but it was never reported elsewhere in the country where subtypes D2, C1, and A1 predominate [45]. Within Arunachal Pradesh, genotype I is restricted to the Idu Mishmi community in the Upper Dibang Valley, where it represents around 20.0% of infections, if we consider that untypable HBV infections are due to genotype I [13].

In the whole of China, a review from Li and coworkers reported the presence of genotype I at a very low rate of 1.7% (n = 20/1148) [21]. However, this rate varies substantially from one large Chinese region to another. Genotype I was absent from the Eastern, Northern, and Northeastern Chinese broad administrative subdivisions where the bulk of the ethnic Han population lives. It represents around 2.0% of HBV strains both in Northwestern China (Xibei, which includes the Shaanxi province, as already mentioned) and in Southwestern China (Xinan, which includes Yunnan and Sichuan, among other administrative entities). There was no further precision concerning the geographical position of HBV genotype I carriers in these areas that cover millions of km^2^. By contrast, genotype I is, according to Li, much more frequent (14.0%) in South Central China (Zhongnan, which includes the Guangxi autonomous region). This surprising figure is, perhaps, somewhat overestimated. In a study conducted in five different locations from Guangxi that included 394 HBsAg carriers, genotype I was found in only 3.4% of the cases. Likewise, a recent study conducted in Yunnan (Southwestern, Xinan) retrieved only 0.3% of genotype I (n = 649 HBsAg carriers) [46].

Overall, genotype I is a minor genotype never dominant in its areas of circulation. This situation explains, probably, why it was discovered so late.

## 8. Clinical Consequences of Infections with Genotype I

The clinical case of genotype I is almost empty, as there are no large series of patients depicting the presentation of infection with this virus available in the literature. Patients are often published as case reports, making difficult the elaboration of a relevant presentation of genotype-I-associated disease [36,37,40,41,47].

Chinese investigators, however, tried to provide a description of the features associated with genotype I infection. Li and coworkers observed an imbalance in the sex ratio of genotype-I-infected habitants of southern Guangxi counties. Although there was no difference in the age of the carriers, the authors observed that HBeAg prevalence was the lowest in genotype-I-infected patients (16.7% versus 31.7% with genotype C). This difference was non-significant, however [48]. In the rural Long’an cohort from Guangxi, genotype I represented more than 90.0% of the initially untypable HBV isolates (n = 38/281). Interestingly, genotype I (or X/C) also represented 22.0% of the 40 patients who developed a hepatocellular carcinoma, while the genotype I proportion was only 12.0% in those who do not develop liver cancer. Despite this difference not reaching a level of significance (*p* = 0.08, ns), it suggests that infection by genotype I is not benign at all [9].

## 9. Genetic Variations Observed on HBV Genotype I Genome

Full sequences (n = 121, Appendix A) retrieved in the database were investigated for the presence of clinically relevant susceptible sequence variations affecting the pre-core region, the basal core promoter, the negative regulatory element, the pre-S region, the major hydrophilic region, or the reverse transcriptase domain of the polymerase gene [49,50]. All significant variations are summarized in Table 2. The examination of the table indicates that the currently available strains of genotype I are, in general, mildly mutated when compared with what we know about the genotype B and C also present in the same region. It is known that relevant genetic variations tend to accumulate in each patient with time and with the progression of liver disease [51,52,53]. The overall low rate of mutation commonly observed in the HBV genotype I genome may have different explanations. Genotype I was frequently sampled during surveys conducted in apparently healthy participants identified as members of ethnic minorities with the aim to investigate the molecular epidemiology of HBV strains [8,46,54]. Thus, patients with liver diseases that have progressed, harboring a potentially heavily mutated version of the HBV genome, were most probably not selected in the surveys, with the notable exception of the one published by ZL Fang and coworkers [9]. Another explanation would be that the mutation rate of HBV genotype I is lower than those of other genotypes, leading to a lower production of quasi-species with escape mutations. Such circumstances, possibly linked to the genotype G origin of the reverse transcriptase domain of the viral polymerase, might allow for a better control of the infection by its host and provide another explanation for the apparently low association between genotype I and severe disease [55]. Of course, this hypothesis deserves more substance and should be confirmed by undertaking well-designed longitudinal or case-control studies. It is supported, however, by the low nucleotide diversity observed by Phan and coworkers [20].

We also have to mention the somewhat unexplained presence of a large subset of genotype I sequences, all coming from Guangxi, which harbor, contrary to other strains, a very high rate of mutations at clinically relevant hotspots. These strains are responsible for the higher mutation rate affecting genotype I in Chinese data (Table 2) and appear as a divergent cluster on a genotype I phylogeny (Figure 6) [56]. A clinical description of diseases associated with this family of sequences is necessary to obtain a better appraisal of their impact on their host.

## 10. Infected Populations

The reading of the literature produced by both the Chinese and Indian investigators indicates that genotype I is a minor HBV genotype infecting primarily ethnic minorities in these two gigantic countries.

In India, genotype I was reported to infect only members of a small and relatively isolated ethnic group (7000 persons), the Idu Mishmi who speak a Tibeto-Burman language and live in a mountain district of the Upper Dibang valley, Arunachal Pradesh [8]. They are related to the Lhoba people, one of the smallest Chinese ethnic minorities living in Southeast Tibet. Chronic infection with HBV is known to be hyperendemic among Idu Mishmi (HBsAg 21.2%). They represent, however, only a droplet in the ocean of 37 million Indian HBsAg carriers [57].

We already mentioned that, in Guangxi, an early publication of Fang and coworkers observed that genotype I represents a particularly high proportion of HBV strains (>13.0%) [9]. Hence, Long’an County, like most of Western Guangxi, is almost exclusively peopled by the Tai-Kadai-language-speaking Zhuang ethnic group, the largest minority of China (18 million) [58]. This same group of authors carried on its exploration and found genotype I in patients from two counties out of five in Guangxi (Bing Yang and Na Po), sometimes in a very high proportion (15.3%) [48]. However, this survey demonstrated that genotype I is not restricted to ethnic minorities, as Bing Yang is almost exclusively peopled by the Han. This latter observation remains marginal in the Chinese context. Other ethnic minorities from Guangxi, the Dong and the Miao, were not infected by genotype I. According to Li and coworkers, the phylogeographic analysis of local genotype I strains, all of the I1 subtype, shows that they find their origin in Long’an county, in the Western part of Guangxi where the Zhuang ethnicity is dominant [48]. Genotype I is less frequent in Guilin (0.7%, n = 2/276 HBsAg carriers), Eastern Guangxi, where a large (>90.0%) majority of Han people are living [59]. In the Sichuan province, located northwest of Guangxi, for example, genotype I was found in ethnically Yi children [54]. The Yi people speak a Tibeto-Burman language and represent another of the 55 minor ethnic groups catalogued in China. There are around 10 million Yi people spread in four countries (China, Thailand, Laos, and Vietnam). They live in the highlands, and Chinese investigators explicitly consider that a significant subset of the Yi lives in isolated villages difficult to reach by public health campaigns, such as universal anti-hepatitis B immunization. Likewise, in Yunnan, in a series of 153 HBsAg(+) sera, all of them sampled in ethnic minorities (Dais, Yaos, and Hanis), a small percentage (2.6%) were positive for genotype I [24]. A more recent survey conducted on 649 members of 20 different ethnic minorities from Yunnan detected only 0.3% (n = 2) of genotype I. Both strains were found in Zhuang participants, supporting the view that this ethnicity is the most impacted by genotype I in China [46]. An examination of the distribution area of genotype I in China suggests that it could probably be found in ethnic minorities of the Guizhou province if it is carefully searched for.

In Laos and Vietnam, the association of genotype I infection with specific ethnic minorities was not reported so far. However, in both countries, genotype I isolates come from patients living in the northern regions known to shelter large proportions of ethnic minorities (including Tibeto-Burman-, Mon-Khmer-, and Miao-Yao-speaking communities) [60,61]. In Laos, nearly half of the population belongs to a minority different from the Lao majority. Moreover, related ethnic communities live on both sides of the Laos–Myanmar or Thailand–Myanmar borders in the neighboring mountain states of Shan and Kachin (that forms a wedge between Yunnan and Arunachal Pradesh), suggesting that genotype I is most probably also present in Myanmar.

Some authors consider that the cradle of genotype I lies in North Vietnam, and that its emergence is recent (XIXth century) and related to the importation of the HBV strains of genotypes A and G during the French colonization of Vietnam. The spread of the virus could have been the consequence of the century-long turmoil generated by the successive wars in Indochina and Vietnam. The same authors hypothesized that the spread of genotype I in China followed that of HIV [24]. This interesting hypothesis does not explain why genotype I is present in Arunachal Pradesh, which escaped the aforementioned conflicts, and is absent from Cambodia or Southern Vietnam, which experienced both colonization and terrible periods of war [23,62], nor does Shen’s hypothesis explain the counterintuitive observation that genotype I preferred to infect ethnically segmented minorities from highlands rather than the ethnic majority living in more densely populated coastal plains. 

We observed that the area of HBV genotype I circulation corresponds greatly to the Southeast Asian Massif (SEAM), a vast region of hills and mountains (an elevation of 500 m above sea level in general) of more than 2.5 million km^2^ [63]. The SEAM is ethnically distinct from the lowlands and is inhabited by dozens of ethnic minorities without a real cultural unity except that these people tried for millennia to resist the rule of empires more solidly established in richer plains [63]. These homogeneous traditions of political autonomy of the local highlanders prompted some scholars to call this region Eastern “Zomia”, a word meaning “people of highlands” in Tibeto-Burman languages (Figure 4) [64,65]. Interestingly, the SEAM/Eastern Zomia includes the Tibetan–Yi corridor, in western Sichuan, a region that played a major role in trade, human migrations, and genetic admixture between Tibet (West) and China (East) for millennia, and was thus suitable for the emergence of recombinant Hepadnaviruses [66]. We hypothesize that the compound structure of genotype I might be the result of the recombination of an ancient/extinct HBV A/G recombinant brought by Western Eurasians such as Neandertals, Denisovans, or the Yamnaya people and transmitted to Asians during a process of genetic admixture with East Asians (Tu people from Qinghai and Gansu, for example, harbor a significant proportion of West Eurasian ancestry in their genome), whereas genotype C should have been brought by Northeastern Asians who began a massive expansion with the domestication of rice and millet in the Yangtze Valley four millennia ago [67,68,69,70,71]. Due to the frequent mother-to-child transmission of HBV in East Asia, we hypothesize that the virus should frequently segregate with the mitochondrial DNA (mtDNA) of mothers [14]. An analysis of the mtDNA of genotype I carriers might provide more solid evidence concerning the origin of the virus.

## 11. Conclusions

Genotype I is a rare HBV genotype endemic to continental Asia and detected in only five countries so far. As an apparently successful recombinant from three different genotypes (G, A, and C), it appeared only recently and well after the eight (A–H) original HBV genotypes. Its exact geographical place of apparition is unknown. Genotype I is only responsible for a minority of HBV infections in its circulation area, where it predominantly affects ethnic minorities living in the Southeast Asian Massif, also called Eastern Zomia by anthropologists. This marginal presence in the politically dominated human populations of some of the largest nations (and, formerly, empires) of the world is plausibly responsible for its late discovery. The way it managed to endure in the realms of epidemiologically more successful HBV genotypes (notably, B and C) is also a kind of epidemiological mystery. The natural history of genotype I infection is poorly documented, and it will, for obvious reasons, probably remain so in the current era of universal anti-hepatitis B immunization at birth.

## Figures and Tables

**Figure 1 microorganisms-11-02204-f001:**
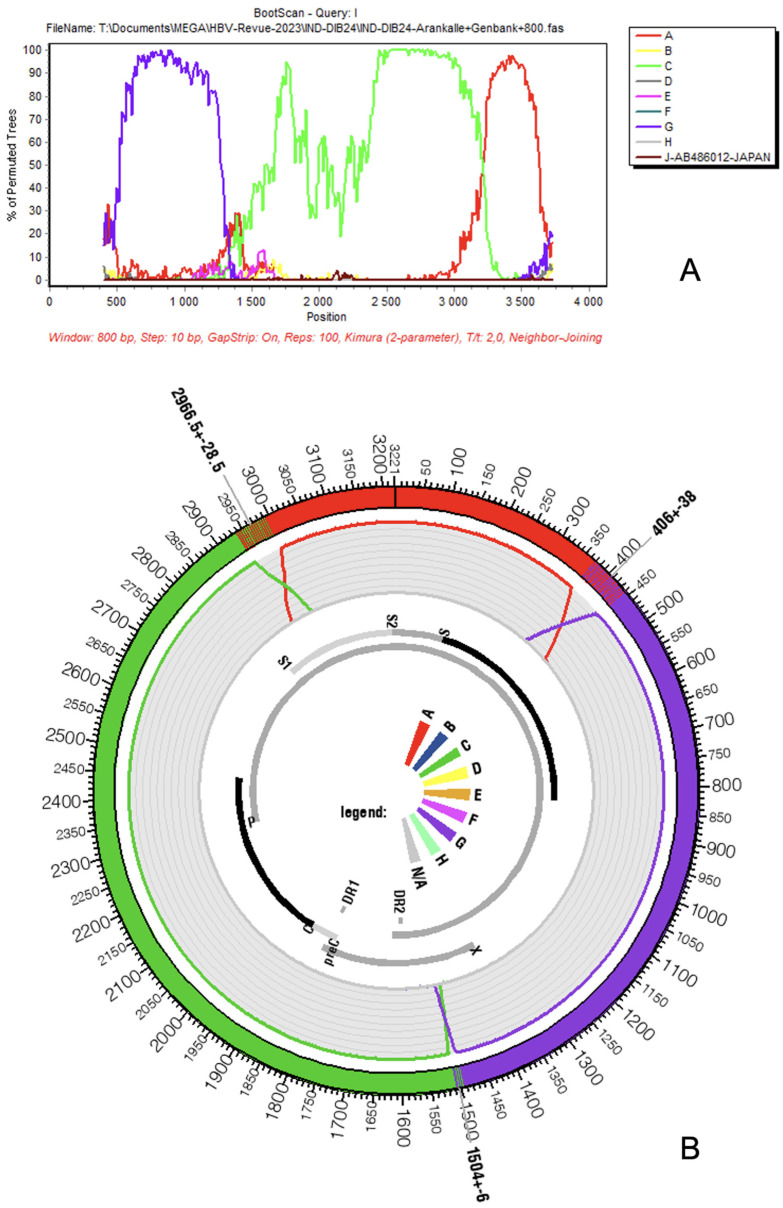
Genetic composition of HBV genotype I according to SimPlot (**A**) and jpHMM (**B**). Colors of original genotypes are conserved in both figures (A: red, C: green, and G: purple). Analysis by SimPlot was not performed using a reference sequence comparison. We preferred, instead, to proceed with a bulk analysis that considers, for each genotype, the set of full-length sequences mentioned on Figure 2. Parameters used were those already employed by [8].

**Figure 2 microorganisms-11-02204-f002:**
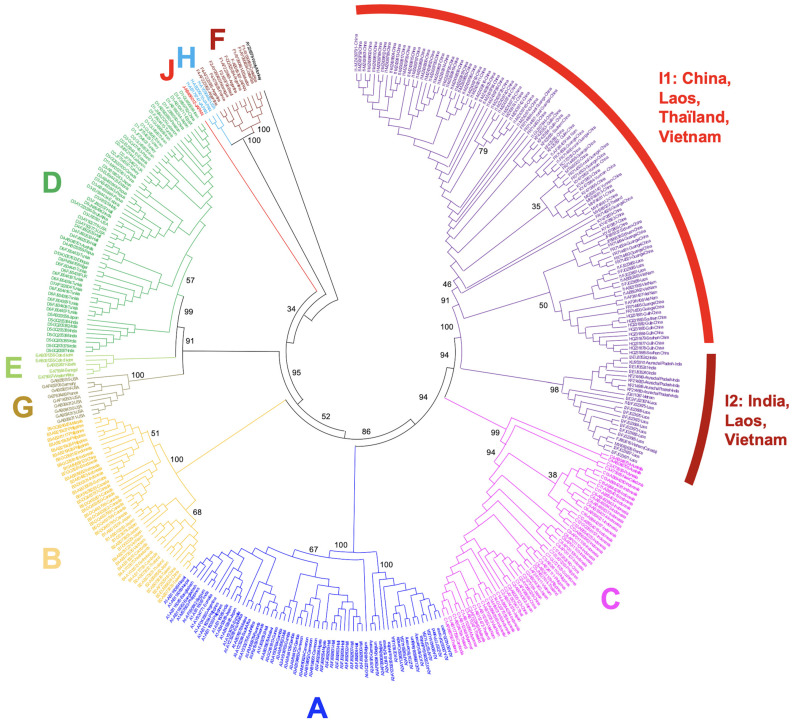
Circle cladogram of HBV genotypes. The two subtypes and genotype I are clearly differentiated. In total, 121 full-length genomes of HBV genotype I have been included. Other HBV genotypes (A–H,J) are differentiated by colors. The circle cladogram was generated on MEGA XI using the maximum likelihood method and Tamura-Nei 2-parameter model with 1000 bootstrap iterations. Bootstrap values are mentioned for the main nodes.

**Figure 3 microorganisms-11-02204-f003:**
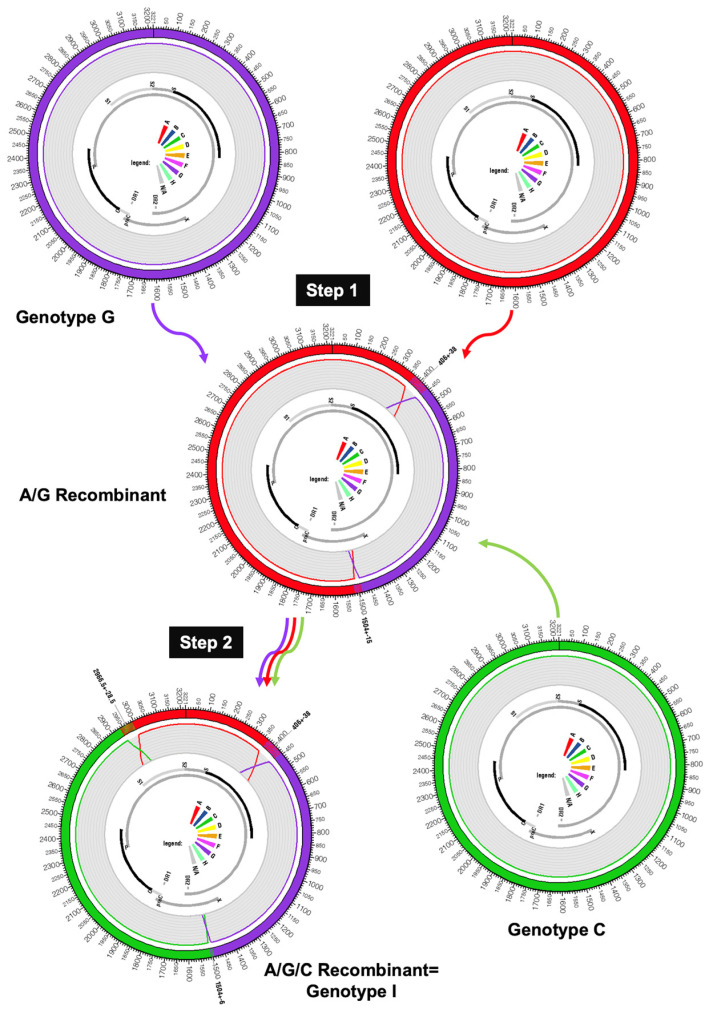
Hypothetical reconstruction of genotype I emergence. The hypothetical first event of recombination associated a genotype A strain and a genotype G strain to remove the non-functional HBeAg of genotype G. Then, a second event of recombination brought the more efficient enhancer 2, basal core promoter, pre-core, and core gene of a genotype C isolate. Color code from Figure 1 is preserved (A: red, C: green, and G: purple).

**Figure 4 microorganisms-11-02204-f004:**
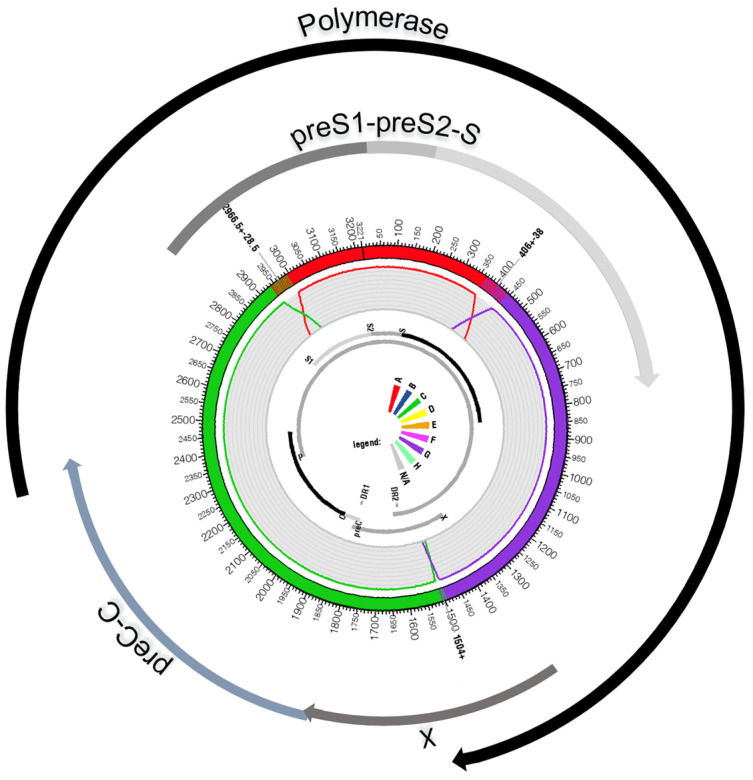
HBV genotype I genome and its open reading frames. The table below recapitulates the composition of each ORF regarding each original genotype that participated in viral genesis.

**Figure 5 microorganisms-11-02204-f005:**
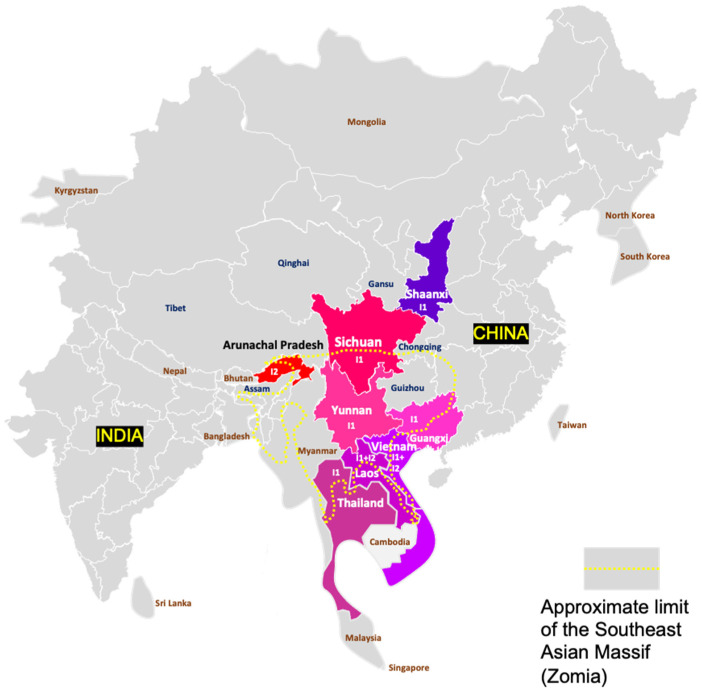
Geographic distribution of genotype I in continental Asia. Independent countries and subnational geographical entities where genotype I has been isolated are colored. Other territories are left in grey. The approximate limits of Southeast Asian Massif, also called Zomia by anthropologists, are materialized by a dotted yellow line. The I subtypes of HBV isolated in a given territory are mentioned.

**Figure 6 microorganisms-11-02204-f006:**
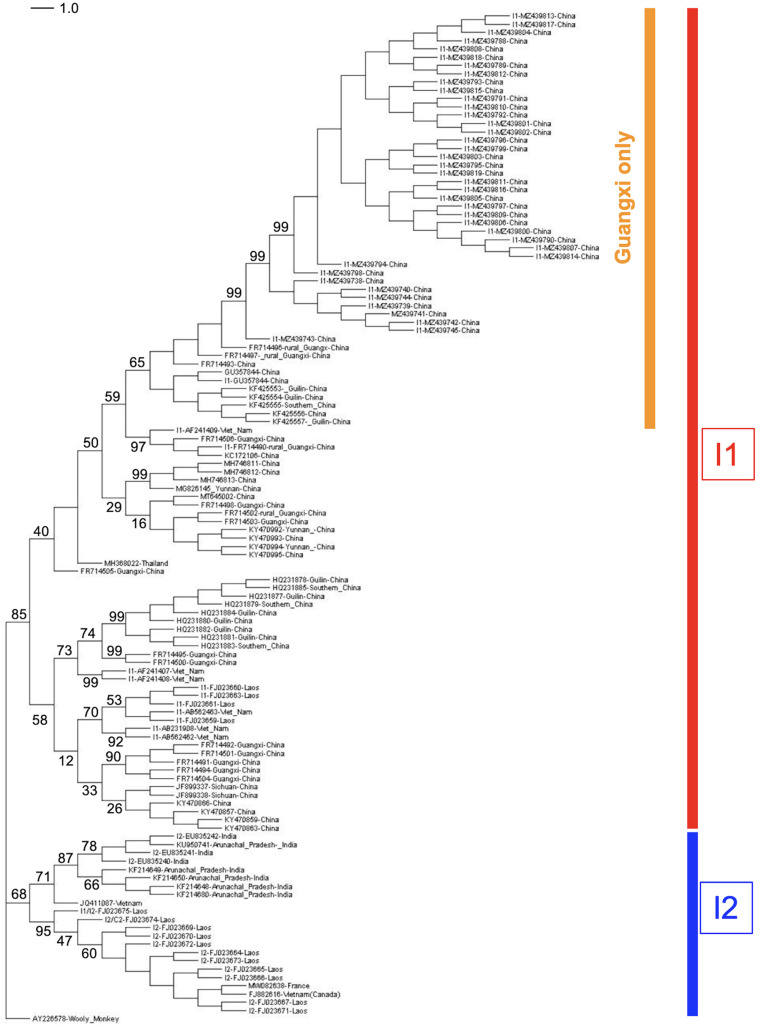
Dendrogram including 121 full sequences of genotype I. Subtypes I1 and I2 are well-differentiated. Within I1 branch appears a large cluster of I1 strains from Guangxi, which form a homogeneous group with much more nucleotide variations than other strains. The tree was generated as in Figure 2 and transformed with Dendroscope-3 to account for more precision of the evolutionary dynamics affecting some genotype I strains. Bootstrap values are mentioned for the main nodes.

**Table 1 microorganisms-11-02204-t001:** Composition of each open reading frame of HBV genotype I according to the genotype of origin. Percentages which do not reach 100% correspond to uncertainties of attribution around recombination points.

Genes	Position on HBV Genome	Genotype A	Genotype C	Genotype G
**Polymerase**	2307-1623	25%	25%	43%
**PreS1**	2848-3204	56%	28%	
**PreS2**	3205-154	100%		
**HBs**	155-835	35%		63%
**PreC-C (HBe-HBc)**	1814-2454		100%	
**HBx**	1374-1838		70%	27%

**Table 2 microorganisms-11-02204-t002:** Main variations of HBV genotype I according to country of origin and subtype. Only full sequences have been considered (n = 121). Number in table cells correspond to the number of times the variation is observed. In parentheses are indicated the non-synonymous change when it does not correspond to one usually described.

Gene or Regulatory Element	Nucleotide or Amino-Acid	Viet Nam (6) I1	Viet Nam (2) I2	Laos (4) I1	Laos (12) I2	China (88) I1	India (8) I2	Thailand (1) I1
**Pol**	rtA181T/V	0	0	0	0	2 (T)	0	0
**Pol**	rtM204I/V/L	0	0	0	0	1	0	0
**Pol**	rtL229V/M/F	1	0	0	0	0	0	0
**Pol**	rtN236T/I	0	0	0	0	1	0	0
**pre-S2**	3174-76, M1V/T/I	0	0	0	0	1 (I)/1 (R)	1 (T)	0
**pre-S2**	T54C/TT53-54CC/T53C, F22S/P/L	0	0	0	0	1 (S)/4 (P)/9 (L)	1	0
**HBs**	C312T, S53L	0	0	0	0	1	0	0
**HBs**	G542A, G130R	0	0	0	0	14 (S)	0	0
**HBs**	G553A, M133T	0	0	0	0	4 (I)	0	0
**HBs**	G587A, G145R	1	0	0	1	0	3	0
**HBs**	G700A, W182Stop	0	0	0	0	6	0	0
**HBs**	T704G, W184A	0	0	0	1	0	0	0
**HBs**	763-765, P203Q	0	0	0	1 (Q)	1(S)	0	0
**HBs**	T766A, S204R	0	0	0	1 (N)	2 (G)	7 (N)/1 (R)	0
**HBs**	T784G, S210R	0	0	0	0	2	0	0
**HBs**	T801A, L216Stop	0	1	0	0	0	0	1
**HBs**	T813G/T814G, F220L/C	1 (L)	0	0	0	40 (C)	0	0
**enhI/II, HBx, Pol.**	G1613A, E80E, R841K	0	1	5	0	48	0	
**enhI/II, HBx**	C1653T, H94Y	0	0	0	0	36	0	0
**enhI/II, HBx**	T1753C, I127Y	0	0	1	0	48	0	1 (G)
**Basal Core Promoter, HBx**	A1762T, K130M	2	0	1	1	56	0	1
**Basal Core Promoter, HBx**	G1764A, V131I	2	0	1	1	56	1	1
**Basal Core Promoter, HBx**	A1762T/G1764A	2	0	1	1	54	0	1
**Basal Core Promoter, HBx**	C1766T, F132Y	0	0	0	0	1	3	0
**HBx**	T1674C, S101P	0	0	0	0	2	1	0
**HBx**	A1727T, A118N	0	0	0	0	1 (C)	0	0
**HBx**	1767-1769, F132Y/I/R	0	0	0	0	1(Y)/1 (I)	2 (Y)	0
**HBx**	C1773T, L134L	0	0	0	0	1/1 (A)	7	0
**HBe**	A1846T, S11S	0	1	0	0	35	1	0
**HBe**	G1896A, W28Stop	0	0	0	3	32	3	1
**HBe**	G1899A, G29D	0	0	0	1	12	1	0
**HBc**	T1961G, S21A	0	0	0	1	2(T)	0	0
**HBc**	A2131T, E77D	0	0	0	1 (D)	3 (D)	1 (Q)	0
**HBc**	2139-40, A80T/I/G	0	0	0	1(S)	1 (T)	0	0
**HBc**	2159-61 S87R	0	0	0	N (1)/G (1)	R (41)	N (1)	1 (G)
**HBc**	A2189T, I97F	0	0	0	1	1	1	0
**HBc**	A2439C, E180A	0	0	0	1 (G)	1 (K)	0	0

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
