# Peer review of "Concealed for a Long Time on the Marches of Empires: Hepatitis B Virus Genotype I"

_microorganisms, 2023, doi:10.3390/microorganisms11092204_

Round 1

Reviewer 1 Report

The review by Agnes Marchio and coauthors is very interesting. This is a very comprehensive analysis of available data on genetics and epidemiology of very particular HBV genotype, namely genotype I. The review is perfectly written and well-structured. I have a few minor comments.

1. The SimPlot (Fig. 1A) has a poor resolution.

2. Lines94-95: Obviously, genotype I shoul not be mentioned among “other genotypes” in parenthesis.

3. Table 1, S protein row: 35% should be indicated for genotype A, not C.

Author Response

Reviewer 1

  1. The Simplot (Fig. 1A) has a poor resolution.

We apologize for this shortcoming. We increased the resolution of the new version of this figure.

  1. Lines 94-95: Obviously, genotype I should not be mentioned among “other genotypes” in parentheses.

The reviewer is right. It should not be there. We removed genotype I from the parentheses.

  1. Table 1, S protein row: 35% should be indicated for genotype A, not C

We thank the reviewer for its acute reading of our work. We corrected the mistake in the new version of Table 1.

Reviewer 2 Report

This review by Marchio et al. provides a comprehensive analysis of the major molecular features of hepatitis B virus (HBV) genotype I and characteristics of infected populations in China, India, Laos, Thailand, and Vietnam. The manuscript presents valuable information on HBV genotype I but requires attention to address several concerns.

Comments:

1.         As a review article, Section 2 of Materials and Methods is unnecessary and should be removed.

2.         The authors mention genotype I in five countries (China, India, Laos, Thailand, and Vietnam) (page 1, lines 23-24; page 13, lines 412). However, Figure 2, Table 2, and Figure 6 lack the genotype I strain(s) from Thailand. If the full-length genomic sequence is unavailable for HBV genotype I strain(s) in Thailand, the word "Thailand" should be eliminated from the relevant statements throughout the manuscript and Figure 2.

3.         The timing of retrieval for full sequences of genotype I from GenBank should be clarified, as over 120 genotype I strains' full-length genomic sequences are presently available. Hence, Figures 2 and 6, along with Table 2, should be updated.

4.         The legends to Figures 1, 2, 5, and 6 are superficial and require elaboration, accompanied by details of the methods used and compared HBV strains.

5.         Figure 1: Genotype J is absent in Figures A and B, necessitating proper labeling. The HBV strain used for Simplot analysis must be specified for each genotype.

6.         Figure 2: The Thailander strain is missing in the I1 cluster, while the Canadian strain is present in the I2 cluster. The authors should explain the reasons for this inconsistency and provide bootstrap values for major nodes to support clear differentiation. Additionally, label genotypes A-H, and J.

7.         Figure 6: This figure includes three I2 strains from European countries (one from France and two from Canada). The authors stated that two patients originating from Vietnam were characterized in France and Canada (page 8, lines 225-226). Was the remaining one isolated from a patient born and living in Canada? If so, the authors must revise their conclusion that genotype I is endemic to continental Asia (page 13, line 411) and responsible only for HBV infections in ethnic minorities living in Southeast Asian Massif or eastern Zomia (page 13, lines 415-417).

8.         Table 2: Strains with L216stop should be included or the "L216stop" row deleted.

9.         Page 1, line 24: Add "Vietnam" after "Laos."

10.     Page 3, line 95: Replace "I" with "H."

11.     Page 4, lines 129-130: Confirm the availability of "(https://sray.med.som.jhmi.edu/SCRoft-ware/SimPlot/)."

12.     Page 6, lines 191-192: Provide relevant reference(s) for this sentence.

13.     Page 8, line 229: Change "a" to "are."

14.     Page 8, line 231: Modify "countries" to "country."

15.     Page 9, line 288: "Supplemental Table 1" is missing.

16.     Page 14, line 442: Complete the reference.

17.     Remove the duplicate "doi" from references 6, 11, 12, 17, 38, and 46.

Minor editing of English language required as represented in the comments to the authors. 

Author Response

Reviewer 2

  1. As a review article, Section 2 of Materials and Methods is unnecessary and should be removed.

We removed this section from the new version of our manuscript.

  1. The authors mention genotype I in five countries (China, India, Laos, Thailand, and Vietnam) (page 1, lines 23-24; page 13, lines 412). However, Figure 2, Table 2, and Figure 6 lack the genotype I strain(s) from Thailand. If the full-length genomic sequence is unavailable for HBV genotype I strain(s) in Thailand, the word "Thailand" should be eliminated from the relevant statements throughout the manuscript and Figure 2.

To reach the consistency rightfully asked for by the reviewer, we included the sequence from Thailand in the figure (MH368022). The number of complete sequences of Genotype I included reaches now n=121.

  1. The timing of retrieval for full sequences of genotype I from GenBank should be clarified, as over 120 genotype I strains’ full-length genomic sequences are presently available. Hence, Figures 2 and 6, along with Table 2, should be updated.

As suggested by the reviewer we included now 121 full-length sequences in our analysis. The Figures 2 and 6 as well as Table 2 have been updated accordingly as required by the Reviewer. We thank the reviewer for this demand as it improves significantly the quality of our work.

  1. The legends to Figures 1, 2, 5, and 6 are superficial and require elaboration, accompanied by details of the methods used and compared HBV strains.

As requested, we are now providing more detailed captions including the procedures employed to generate figures 1, 2, 5, and 6.

  1. Figure 1: Genotype J is absent in Figures A and B, necessitating proper labeling. The HBV strain used for Simplot analysis must be specified for each genotype.

The full-length sequence of genotype J (AB486012) was introduced in the SimPlot analysis. Analysis by SimPlot was not performed using a reference sequence comparison. We preferred instead to proceed with a bulk analysis that considers for each genotype the set of full-length sequences mentioned on Figure 2. Genotype J could not be used with jpHMM software as this tool is using its own reference strains.

  1. Figure 2: The Thailand strain is missing in the I1 cluster, while the Canadian strain is present in the I2 cluster. The authors should explain the reasons for this inconsistency and provide bootstrap values for major nodes to support clear differentiation. Additionally, label genotypes A-H, and J.

We included the Thai strain in the revised version of Figure 2. Other genotypes are now labelled. Bootstrap values are now provided for major nodes.

  1. Figure 6: This figure includes three I2 strains from European countries (one from France and two from Canada). The authors stated that two patients originating from Vietnam were characterized in France and Canada (page 8, lines 225-226). Was the remaining one isolated from a patient born and living in Canada? If so, the authors must revise their conclusion that genotype I is endemic to continental Asia (page 13, line 411) and responsible only for HBV infections in ethnic minorities living in Southeast Asian Massif or eastern Zomia (page 13, lines 415-417).

To our knowledge patients infected by HBV genotype I characterized in France and Canada were both born in Vietnam. We thus consider that our conclusions concerning the endemic nature of genotype I in ethnic minorities of Eastern Zomia is still valid.

In the future, it is possible that genotype I will be sporadically found in other countries located on other continents. We consider that such observations will not change the endemic character of genotype I to Eastern Zomia (endemic. Of a disease: regularly occurring within an area or community, Of an area: in which a particular disease is regularly found). In our opinion, it will take a very long time (if its ever happens) before this genotype spreads and becomes endemic out of its original geographic cradle. Only massive migrations of populations out of the southeast Asian massif might change genotype I distribution worldwide.

  1. Table 2: Strains with L216stop should be included or the "L216stop" row deleted.

We apologize for this oversight. Two strains harbor (one I1, one I2) this mutation. It is now in Table 2.

  1. Page 1, line 24: Add "Vietnam" after "Laos."

The reviewer is right. We corrected this oversight.

  1. Page 3, line 95: Replace "I" with "H."

We removed genotype “I” and replaced it by “H” in the parentheses.

  1. Page 4, lines 129-130: Confirm the availability of "(https://sray.med.som.jhmi.edu/SCRoft-ware/SimPlot/)."

We are now clearly stating in the text the procedure to get access to SimPlot and jpHMM. SimpPlot is downloadable for free after request to Dr. Stuart Ray (version 3.5.1. Dept of Medicine, Johns Hopkins University School of Medicine, email: [email protected]) while jpHMM (http://jphmm.gobics.de/) is freely available from the indicated webpage.

  1. Page 6, lines 191-192: Provide relevant reference(s) for this sentence.

The statement regarding the nature of C subtypes homology with genotype I stems from our observation. We compared the segment of genotype I putatively coming from genotype C (»1604-2966) with different subtypes of this genotype using Simplot or MEGA tools. Subtypes C6 and C12 were displaying a better homology with genotype I than subtypes C1, C2, or C5 while subtypes C3 and C4 were occupying an intermediate position (see below an illustration with Simplot).

  1. Page 8, line 229: Change "a" to "are."

We corrected this mistake in the legend of Figure 5

  1. Page 8, line 231: Modify "countries" to "country."

We corrected this error.

  1. Page 9, line 288: "Supplemental Table 1" is missing.

We uploaded the Supplemental Table 1 in the revised version of our work.

  1. Page 14, line 442: Complete the reference.

We completed reference 1 with its DOI (doi: 10.3390/genes9100495).

  1. Remove the duplicate "doi" from references 6, 11, 12, 17, 38, and 46.

We apologize for this mistake. All duplicated “doi” were removed as required.

Round 2

Reviewer 2 Report

The manuscript has been revised properly in accordance with my previous comments and improved extensively.